# Rare Earth Elements in *Boletus edulis* (King Bolete) Mushrooms from Lowland and Montane Areas in Poland

**DOI:** 10.3390/ijerph19158948

**Published:** 2022-07-22

**Authors:** Jerzy Falandysz, Innocent Chidi Nnorom, Małgorzata Mędyk

**Affiliations:** 1Department of Toxicology, Faculty of Pharmacy, Medical University of Lodz, 1 Muszyńskiego Street, 90-151 Łódź, Poland; 2Analytical/Environmental Unit, Department of Pure and Industrial Chemistry, Abia State University, Uturu P.M.B. 2000, Nigeria; innocent.nnorom@abiastateuniversity.edu.ng; 3Laboratory of Environmental Chemistry and Ecotoxicology, University of Gdańsk, 80-309 Gdańsk, Poland; gosiamedyk1@wp.pl

**Keywords:** fungi, human, REEs, trace elements, scandium, yttrium, lanthanides, forest, wild food

## Abstract

Mining/exploitation and commercial applications of the rare-earth elements (REEs: La, Ce, Pr, Nd, Sm, Eu, Gd, Tb, Dy, Ho, Er, Tm, Yb, and Lu) in the past 3 decades have raised concerns about their emissions to the environment, possible accumulation in food webs, and occupational/environmental health effects. The occurrence and distribution of REEs Y and Sc in the fruitbodies of *Boletus edulis* collected from geographically diverse regions in Poland were studied in 14 composite samples that were derived from 261 whole fruiting bodies. Individual REE median concentrations ranged from 0.4–95 µg kg^−1^ dry weight (dw). The summed REE concentrations varied widely, with a median value of 310 µg kg^−1^ dw and a range of 87 to 758 µg kg^−1^. The Sc and Y median concentrations (dw) were 35 and 42 µg kg^−1^, respectively. Ce, La, and Nd, with median values of 95, 51, and 32 µg kg^−1^, respectively, showed the highest occurrence. *B. edulis* collected from a forested area formerly used as a military shooting range—possibly a historically contaminated site—had an elevated summed REE content of 1796 µg kg^−1^. REE concentrations were generally low in Polish King Bolete. Dietary intake from a mushroom meal was negligible, posing no health risk to consumers.

## 1. Introduction

The term rare-earth elements (REEs) includes La, Ce, Pr, Nd, Sm, Eu, Gd, Tb, Dy, Ho, Er, Tm, Yb, and Lu—all of the lanthanides series [1]. Scandium (Sc) and yttrium (Y) are also studied in foods and environmental materials and considered together with the REEs [2,3,4]. The REEs, Sc and Y occur in soil at relatively low concentrations and their mineral deposits worldwide are limited, thus they are considered critical metals [5,6]. Nowadays, REEs are a highly sought after resource for industrial technology and numerous other applications [7]. Mining and processing of the REE ores like that of some other metallic elements often result in pollution of underground waters, the pedosphere, and atmosphere as well as in elevated levels in vegetation at least at the local or regional level [8,9,10].

In an early study by Ichihashi and co-workers (1992), the herb pokeweed (*Phytolacca americana*) and two species of ferns (*Dicranopteris dichotoma* and *Athyrium yokoscence*) were observed to be good accumulators of REE Sc and Y compared to other species investigated, while the species *Ph. americana* closely reflected their pattern in the soils [11]. Concentration of REEs in plants can be higher than is typical if they grew in polluted soil [3,6,12]. 

From the food, environmental, and toxicological point of view, the intentional or unintentional production, release, exhaust, or other emissions of REEs pose a risk to the health of humans and other biota. The particle size of the emission/release may determine the deposition/exposure, and this may be more harmful if they are nanoparticles [13]. Apart from the pedosphere, other sources of REEs for leafy vegetation are fallout from atmospheric dust and nanoparticles with REEs from industrial and commercial emissions [14]. REEs are considered to have possible toxicological effects on humans in occupational and environmental settings, though toxicological data are scarce [5,15,16,17,18]. Long-term exposure to dust containing REEs can cause lung disease [19], an effect that is similar to other inorganic dusts under occupational exposure. 

Apart from occupational REE exposures and the associated toxicity, the environmental occurrence of REEs and their growing production, use, recycling, and emissions elicits interest regarding their possible accumulation in terrestrial food webs and the potential risks presented from consuming such contaminated foods. Wild mushrooms are traditional food items in many regions of the world and parts of terrestrial food webs. They are actively traded internationally. Dried mushrooms are known to be rich in minerals, and it is a difficult task to construct a reliable baseline dataset for such a biodiverse food. Some of the challenges to generating such a dataset include limited knowledge of (i) the mushroom species’ bioconcentration potential for minerals from different geochemical backgrounds, (ii) the role of geogenic and anthropogenic factors, (iii) biochemical pathways of mineral biotransformation and forms (speciation), as well as (iv) the effect of processing (and cooking), accessibility, and bioavailability from mushroom meals [20]. In recent years, apart from other foods, there is a growing interest in the occurrence, accumulation, and intake of a wide range of REEs from wild, edible mushrooms. In the first published set of analytical data on 14 REE elements (14REE) in mushrooms, some differences in concentration between species, as well as differences in occurrence between specimens of the same species, could also be seen when determined by sectoral field mass spectrometry [21].

Borovička et al. critically reviewed the literature on REEs in mushrooms available up to 2010, while recently Zocher et al. conducted a similar study and included the literature for the period up to 2018 [22,23]. Both reports highlighted some biased results of high REE concentrations in the published literature. Contamination of mushrooms with soil debris causes false positive results (elevated REE concentrations) as does a poor choice of analytical methodology, and especially instrumentation, for REE measurements [22]. Critical methodological parameters include avoiding spectral interferences, high resolution and sufficient method sensitivity, and, sometimes, a need for pre-concentration due to ultra-trace occurrence [23]. Consequently, the amount of reliable data on REE levels in mushrooms is considered to be very small—and only available for relatively few species of fungi or only for one specimen in some cases [4,21,22,23,24,25,26]. In the abovementioned studies, it has not yet been shown that any of the species studied accumulated REEs particularly strongly, although there were differences in the REE contents between mushrooms of the same species that were collected from different, geographically distant stands.

*Boletus edulis* Bull., is an indigenous and edible mycorrhizal mushroom (King Bolete) that is relatively popular and widespread in Europe and in the northern regions of Asia. This species is relatively expensive and widely known in the gourmet tradition in Europe [27]. Both fresh and preserved *B. edulis* (including dried products) are widely traded locally/regionally and even internationally. The European *B. edulis* is well characterized in the context of some nutritionally essential metallic macro- and micro-nutrients such as K, Mg, Cu, Zn, and Se as well as toxic and radiotoxic elements (Ag, As, Cd, Hg, methylmercury, Pb, ^134/137^Cs), some of which accumulate in fruiting bodies, but data is largely lacking for REE contents [22,28,29,30,31,32,33,34,35,36,37]. Some early data on REE levels in a pool of five caps of *B. edulis* collected from the northern region of Poland were published by Falandysz and co-workers [21]. In this study we examined the status of 14 individual REEs and also Sc and Y in morphological parts of the fruiting bodies of *B. edulis* collected from spatially distant locations in Poland. This paper also examined the shale-normalized distribution of REEs in *B. edulis*. Plotting the REEs or REEY fractionation pattern helps to understand the relative changes in their abundance due to natural geochemical processes and anthropogenic (enrichment) influences. Since results were obtained for REEs in raw mushrooms, a rough estimate of the possible intake of REEs from bolete mushrooms in a prepared meal was also calculated. 

## 2. Materials and Methods

The *B. edulis* samples were collected over a wide area, from 15 locations in northern and southern Poland, during the mushrooming season (usually early September) between 1998 and 2008 (Figure 1). Poland has a largely flat landscape with uplands in the south, and agricultural land and forests dominate, and it has four seasons—spring, summer, autumn, and winter. The climate (temperature and meteorological conditions) is moderate and changing and becomes more continental with increasing distance from the sea to the montane south of the country. The mean annual temperature of around 7–8 °C in the north increases slightly towards the south. Yearly precipitation is around 550 to 600 mm and slightly higher in the uplands than in the plains, with up to 1300 mm regionally in the montane areas. Soil bedrock in Poland is highly diverse and is reflected in the soil types with a predominance of brown earth (brown and lessive soils), podzols (podzolic soils, podzols, and rusty soils), and several other types [38].

Mushrooms used in this study were specimens from a repository [36]. Before storage in dry and clean conditions, the fungal materials were properly cleaned, dried, and ground after collection and packed in tightly closed plastic containers. Mushrooms that are processed and stored in these conditions are considered to be valid materials for the study of minerals and trace elements decades after collection [39]. To gain insight into the distribution of REEs, Sc and Y between the two major morphological parts of the fruiting body, the specimens from some locations were separated into cap (or pileus with skin) and stipe (or stem). Depending on the number collected at each site, between 10 and 45 specimens were pooled into composites by location (Figure 1; Table 1) [40]. 

To determine the REEs, Sc and Y, circa 200 mg quantities of finely powdered composites of the fungal materials were mixed with 3 mL solution of ultrapure concentrated nitric acid (HNO_3_, 65%) and 1 mL of ultrapure hydrofluoric acid in a polytetrafluoroethylene (PTFE) tube. Then, the tubes were screw-tightened in a stainless steel jacket and digested in an oven at 150 °C for 78 h. The resulting solutions obtained were evaporated to dryness at 110 °C to remove the excess HF, dissolved in about 1 mL of HNO_3_ in a clean sample tube, and diluted to a final volume of 50 mL with ultrapure water. As an internal standard, rhodium (Rh) (10–20 micrograms per litre) was added to each sample prior to the analysis by inductively coupled plasma mass spectrometry (ICP-MS, ELAN DRC-e, PerkinElmer Inc., Woodbridge, ON, Canada) [24,41].

In order to achieve good analytical quality control and quality assurance, blanks and two certified reference materials (CRMs) were examined. Each element was measured 3 times and the values of relative standard deviation (RSD) were within 5% in the samples and the certified values for the CRM. Due to unavailability of a REE CRM for mushrooms, citrus leaf (GBW 10020) and soil (GBW 07405) CRMs produced by the Institute of Geophysical and Geochemical Exploration, China, were used [40,42]. The limit of quantification of the method for individual elements was in the range from 0.1 ng g^−1^ to 2 ng g^−1^. The results obtained for *B. edulis* were further normalized against REEY patterns in certain shales [25,43].

REE intake through mushroom consumption was estimated using the following equations [44]:EDI = (Cm × CR)/BW (1)
where EDI is the estimated daily intake of REEs from mushroom consumption; Cm is the ∑REE in the mushroom (wet weight); CR is the consumption rate [100 g wet (whole) weight for normal consumers and 300 g wet (whole) weight for high level consumers]; and BW is the body weight (60 kg for adults and 30 kg for children). The intake dose of REEs that was found to be damaging to human health was reported to be 100–110 µg kg^−1^ BW day^−1^, which was certificated through a human health study in REEs mining areas and from animal experiment results [3,45,46].

## 3. Results and Discussion

### 3.1. REE Concentration Levels

The range of values for each REE, Sc and Y, determined in *B. edulis* are listed by morphological part of the fruiting body and place of collection in Table 1. The median values of the individual elements derived for whole fruiting bodies, caps, and stipes in this study were in the range (in µg kg^−1^ dw), 5.9 to 35 for Sc, 31 to 43 for Y, 33 to 56 for La, 60 to 110 for Ce, 6.4 to 11 for Pr, 22 to 41 for Nd, 3.4 to 6.3 for Sm, 0.6 to 1.1 for Eu, 4.3 to 5.2 for Gd, 0.7 to 1.2 for Tb, 3.8 to 5.5 for Dy, 0.7 to 1.1 for Ho, 2.1 to 3.6 for Er, 0.4 to 0.6 for Tm, 2.0 to 3.5 for Yb, and 0.4 to 0.5 for Lu (Table 1). These median values for the REEs as well as other data obtained for almost all locations in this study agree very well with an earlier report on the concentrations of lanthanides (using sector field mass spectrometry) accumulated in a few *B. edulis* caps collected at some locations from the northern region of Poland [21].

Macromycetes take up from soil substrata both organics and minerals and accumulate these in the mycelium, and mycorrhizal species also share some nutrients and water with symbiotic plants. They are then gradually translocated to the fruiting bodies based on needs—this is believed to be largely regulated in the case of the macronutrients (K, P, S, Mg) and major micronutrients (Zn, Cu, Ca, Na, Se). The co-accumulation of some other elements alongside those listed above also follows the same physiological pathways, often because of physical and chemical similarities between the elements, though some particular mushroom species show specific ability for accumulation of certain elements, e.g., V (some species need it), Ag, Hg, etc. Despite this, soil geogenic signature and especially anthropogenic pollution (As, Cd, Hg, Pb, Sb, ^137^Cs, ^90^Sr, etc.) also contribute much to the uptake and accumulation of elements by mushrooms. This is observed in studies showing that mushrooms harvested from significantly polluted sites show higher levels of such elements—above the values typical for their species [20,47].

The stipes of fruiting bodies in the present study usually showed higher concentrations of REEs than the caps and the median values of the sum of the REEs (∑REE) at four sites were 359 and 186 µg kg^−1^ dw, respectively (Table 1). The observed higher values of the ∑REE in the stipes compared to the caps of the fruiting bodies of *B. edulis* can be explained by considering the predilection of these elements to Ca [48]. Stipes in mature fruiting bodies are usually richer in Ca. On the other hand, because of their similar chemical and physical properties, REEs are supposed to be absorbed by organisms (e.g., mushrooms) as a group [49]. Divalent Ca, which readily co-occurs in organisms with REEs, occurs at much higher levels in mushrooms, around three to four orders of magnitude, than REEs. For example, studies have reported mean Ca values in caps of *B. edulis* at 39,000 ± 20,000 µg kg^−1^ dw (*n* = 4), while another study reported median values in the range of 29,000 to 170,000 µg kg^−1^ dw for caps and from 54,000 to 250,000 µg kg^−1^ dw for stipes (*n* = 144) [40,50]. 

Available results for REEs (Table 1) and Ca levels in the sample sets of *B. edulis* [21,40,50] indicate a positive correlation in co-occurrence, but observations are too few to examine regression. For example, Ca in *B. edulis* from the Sudety Mts. site had a median Ca concentration level of 160 mg kg^−1^ dw in stipes and 83 mg kg^−1^ dw in caps (id 15; Table 1), and for mushrooms from the Giżycko site, the Ca concentration was 110 mg kg^−1^ dw in stipes and 91 mg kg^−1^ dw in caps (id 10; Table 1) [40,50]. Some similarity in the distribution of REEs and Ca was also found in the mushroom *Lactarius pubescens* [26]. In *Macrolepiota procera* (Field Parasol or Parasol Mushroom) REE levels correlated with Ca and also with Na and Mg [4].

Macrofungi are rather poor Ca bio-concentrators and this is even worse for REEs. For example, the Ca bioconcentration factor (BCF; otherwise quotient from concentration in mushroom and concentration in soil) for a large collection of *B. edulis* ranged from 0.08 to 1.5, and for *Leccinum scabrum* (Rough-stemmed Bolete) from 0.003 to 0.027) [39,51].

**Table 1 ijerph-19-08948-t001:** Concentration levels of the REE, Sc and Y in the morphological parts of *B. edulis* (µg kg^−1^ dw).

Place, Year, Quantity of Specimens and Morphological Part *	La	Ce	Pr	Nd	Sm	Eu	Gd	Tb	Dy	Ho	Er	Tm	Yb	Lu	14REE	Sc	Y
(1) ^¶^ Seacoast Landscape Park, 2007 (15; c)	33	60	6.4	23	4.5	1.1	5.9	0.85	4.8	1.0	2.9	0.5	4.3	0.7	199.95	12	39
(10) Mazury, Giżycko, 2001 (15; c)	73	140	14	42	6.0	1.5	5.3	1.0	5.2	1.1	2.9	0.7	3.5	0.6	348.8	13	39
(11) Mazury, Piska Forest, 2002 (15; c)	15	27	2.3	9.9	1.5	0.5	1.7	0.3	1.3	0.3	0.7	0.2	0.6	0.2	74.3	0.8	12
(14) Tatra Mountains, 1999 (12; c)	14	30	2.9	10	1.8	0.1	1.7	0.3	1.3	0.3	0.6	0.2	1.1	0.1	128.4	46	18
(15) Sudety Mountains., Kłodzka Dale, 2000 (15; c)	35	70	6.7	22	3.4	0.6	4.3	0.7	3.8	0.7	2.1	0.4	2.0	0.4	185.7	2.6	31
Mean ± SD, (caps 5(72)) ^#^	34 ± 24	65 ± 46	6.5 ± 4.7	21 ± 13	3.4 ± 1.9	0.8 ± 0.5	3.8 ± 2.0	0.6 ± 0.3	3.3 ± 1.9	0.7 ± 0.4	1.8 ± 1.1	0.4 ± 0.2	2.3 ± 1.6	0.4 ± 0.3	187 ± 103	15 ± 18	28 ± 12
Median	33	60	6.4	22	3.4	0.6	4.3	0.7	3.8	0.7	2.1	0.4	2.0	0.4	186	13	31
Range	14–73	27–140	2.3–14	9.9–42	1.5–6.0	0.1–1.5	1.7–5.9	0.3–1.0	1.3–5.2	0.3–1.1	0.6–2.9	0.2–0.7	0.6–4.3	0.1–0.7	74–349	0.8–46	12–39
(1) Seacoast Landscape Park, 2007 (15; s)	64	130	13	46	8.4	2.2	9.7	1.5	11	2.4	8.1	1.4	9.8	1.4	433.9	40	85
(10) Mazury land, Giżycko, 2001 (15; s)	110	220	21	64	6.0	0.8	5.0	0.9	4.3	1.0	2.3	0.3	2.7	0.4	477.6	3.9	35
(11) Mazury land, Piska Forest, 2002 (15; s)	21	40	3.7	15	1.3	0.6	1.5	0.3	1.6	0.3	1.1	0.1	0.9	0.1	102.6	1.0	15
(15) Sudety Mts., Kłodzka Dale, 2000 (15; s)	49	97	10	36	6.6	1.1	5.4	0.9	6.7	1.3	5.0	0.8	4.4	0.6	283.7	7.9	51
Mean ± SD (stipes, 4(60) ^#^	61 ± 37	120 ± 75	12 ± 7	40 ± 20	5.6 ± 3.0	1.2 ± 0.7	5.4 ± 3.0	0.9 ± 0.5	5.9 ± 4.0	1.3 ± 0.9	4.1 ± 3.1	0.7 ± 0.6	4.5 ± 3.8	0.6 ± 0.6	324 ± 170	13 ± 16	47 ± 30
Median	56	110	11	41	6.3	0.9	5.2	1.2	5.5	1.1	3.6	0.5	3.5	0.5	359	5.9	43
Range	21–110	40–220	3.7–21	15–64	1.3–8.4	0.6–2.2	1.5–9.7	0.3–1.5	1.6–11	0.3–2.4	1.1–8.1	0.1–1.4	0.9–9.8	0.1–1.4	103–478	1.0–40	15–85
(1) Seacoast Landscape Park, 2007 (15; w)	48	95	9.7	34	6.4	1.6	7.8	1.2	3.9	1.7	5.5	0.9	7.0	1.0	311.7	26	62
(2) Tricity Landscape Park, Osowa, 2006 (10; w)	17	35	3.2	11	1.4	0.4	2.3	0.27	1.5	0.2	0.6	0.1	0.8	0.1	116.8	30	13
(3) Pomerania, Szczecinek, 2000 (22; w)	47	92	8.5	29	4.1	1.3	4.0	0.9	4.8	0.7	2.5	0.6	3.1	0.4	335.9	98	39
(4) Wdzydze Landscape Park, 1999–2001 (45; w)	37	74	7.1	26	4.6	0.8	4.0	0.6	5.4	1.0	3.2	0.5	2.9	0.5	251.6	39	45
(5) Tuchola Pinewoods, 2000 (15; w)	69	140	12	41	7.5	0.9	5.7	1.2	6.3	1.4	3.6	0.6	4.1	0.6	398.9	52	53
(6) Augustów Primeval Forest, 2000 (16; w)	150	270	26	79	10	3.4	8.4	1.4	7.2	1.6	5.4	0.6	4.3	0.7	758	120	70
(7) Warmia land, Puchałowo, 2001 (15; w)	53	95	9.6	25	3.5	1.0	5.0	0.6	4.1	1.0	1.4	0.3	1.8	0.3	235.5	8.9	25
(8) Warmia, Morąg, 1998 (30; w) ^§^	300	630	63	240	44	9.7	36	6.2	33	7.3	21	3.5	24	3.6	1796.3	65	310
(9) Warmia, Olsztyn, 1999 (19; w)	58	110	11	45	6.8	1.2	6.6	1.0	5.5	1.3	3.7	0.5	3.9	0.6	310.1	14	41
(10) Mazury, Giżycko, 2001 (15; w)	91	180	17	53	6.0	1.1	5.1	0.9	4.7	1.0	2.6	0.5	3.1	0.5	470.5	67	37
(11) Mazury, Piska Forest, 2002 (15; w)	18	33	3.0	12	1.4	0.5	1.6	0.3	1.4	0.3	0.9	0.1	0.7	0.1	86.7	0.4	13
(12) Kujawy, Toruń forests, 1999 (16; w)	96	180	19	63	13	2.2	13	2.0	13	3.1	11	1.5	11	1.6	565.4	26	110
(13) Greater Poland, Porażyn, 2008 (13; w)	24	41	3.6	14	2.3	0.5	2.2	0.4	2.0	0.3	1.7	0.2	1.4	0.2	174.8	61	20
(15) Sudety Mountains, Kłodzka Dale, 2000 (15; w)	42	83	8.3	29	5.0	0.8	4.8	0.8	5.2	1.0	3.5	0.6	3.2	0.5	234.9	5.2	42
Mean ± SD (whole fruiting bodies, 14(261) ^#^	75 ± 74	150 ± 150	14 ± 15	50 ± 58	8.3 ± 11.0	1.8 ± 2.4	7.6 ± 8.7	1.3 ± 1.5	7.0 ± 8.0	1.6 ± 1.8	4.8 ± 5.4	0.8 ± 0.9	5.1 ± 6.1	0.8 ± 0.9	327 ± 186	44 ± 36	63 ± 76
Median	51	95	9.7	32	5.5	1.1	5.1	0.9	5.0	1.0	3.4	0.6	3.2	0.5	310	35	42
Range	17–300	33–630	3.0–63	11–240	1.4–44	0.4–9.7	1.6–36	0.27–6.5	1.4–33	0.2–7.3	0.6–21	0.1–3.5	0.7–24	0.1–3.6	86.7–758	0.4–120	13–310

Notes: ^¶^ (Sampling site id, see Figure 1); * c, s, w (caps, stipes, whole fruiting bodies, respectively), TLP (Trójmiejski Landscape Park), WLP (Wdzydze Landscape Park), ^#^ Number of batches and total number of specimens; ^§^ (results for the Warmia—Morąg site “id 8” were not included in calculation of the mean and median values).

The little available data for REEs (such as La and Ce) in mushrooms (*Amanita pantherina*, *Lactarius hatsudake*, *Russula mariae*, *Suillus granulatus*, and *Tricholoma flavovirens*) showed BCF values in the range from 0.003 to 0.027 (La) and from 0.0003 to 0.025 (Ce) [52].

Cerium was the most abundant REE determined in *B. edulis*. Mushrooms collected from the Morąg site in the Warmia region showed an order of magnitude greater quantities of Y, La, Ce, Nd, Sm, Eu, Gd, Tb, Dy, Ho, Er, Tm, Yb, and Lu than specimens from other sites (Table 1). If we exclude any cross-contamination with incrusted sand particles, no other reason can be attributed to this observation other than to associate it with the legacy of an abandoned military area overgrown with forest near the town of Morąg, where mushrooms were collected. *B. edulis* (and some other species from the Morąg site) relative to other locations also showed higher concentrations of other metallic elements (Ba, Co, Ga, Ge, Hf, Hg, Li, Nb, Pb, Sr, Ti, U, V, and Zr) [36], though the mushroom *Amanita muscaria* was not studied for Ba, Ga, Pb, or Sr (and the REE) and *Paxillus involutus* had only data for Hg [53,54]. 

Aruguete et al. earlier discussed the possible impact of geochemical background composition on the abundance of REEs in mushrooms. In their study, differences were observed in the uptake and sequestration of La, Ce, and Nd in fruiting bodies by edible *Amanita rubescens* and *Amanita flavorubescens* collected from different locations [55]. 

Apart from the Morąg site, the next pool of *B. edulis* with elevated level of REEs—when compared to other sites in this study—was from the Augustów Primeval Forest site (id 6) with 758 µg kg^−1^ dw (Table 1). Interestingly, a similar situation was observed for the mushrooms, *M. procera* and *Cantharellus cibarius* (Yellow Chanterelle), from the Augustów Primeval Forest site, which had REE levels well above the median values for these mushroom species collected from other sites across Poland [24]. 

### 3.2. LREE and HREE in B. edulis

The light REEs (LREE; here: La, Ce, Pr, and Nd) occurred in mushrooms in higher concentrations when compared to the middle REEs (Sm to Dy) and heavy REEs (from Ho to Lu) (Table 1). Their median concentrations in the caps, stipes, and whole fruiting bodies followed the same scheme: LREE > MREE > HREE. In detail, the median concentrations (µg kg^−1^ dw) for these subgroups were 121.4, 12.8, and 5.6 in the caps, 218, 19.1, and 9.2 in the stipes, and 187.7, 17.6, and 8.7 in the whole fruiting bodies. The quotient of LREE/HREE was similar for the morphological parts of *B. edulis*, i.e., 21.7 for the caps, 23.7 for the stipes, and 21.5 for the whole fruiting bodies. In work by Bau et al., the quotient of LREE/HREE for a single *B. edulis* specimen (collected in Germany) studied there was 19.6 [25]. Aruguete et al. reported a wide range of concentrations for Nd, La, and Ce between specimens of the same species (*Amanita flavorubescens* and *A. rubescens*), with Nd differing from one to three orders of magnitude, and La and Ce up to two orders of magnitude [55]. 

Hence, the number of individual mushroom specimens collected and tested onsite or contained in a composite sample/pool (sample size) remains essential in characterizing the ability of fungi to bioconcentrate REEs and other minerals and the extent to which they are enriched in the fruiting bodies.

### 3.3. Shale Normalized Patterns of REEY in B. edulis

Figure 2 displays the North American Shale Composite (NASC) and Post—Archean Australian Shales (PAAS) and the European Shale (EUS) and World Shale (WSH) REE and Y (REEY) normalized (normal and log-normal) patterns in the caps, stipes, and the whole fruiting bodies of *B. edulis* based on the median concentration values as well as results for the Augustów (id6) and Morąg (id8) sites and for a specimen examined/reported by Bau et al. [25,43]. Details of the data, including the REE-normalized patterns are presented in the Appendix A. These normalized distributions of REEY in samples of *B. edulis* showed minor differences in patterns between some of the sites. REE Sc and Y, as non-essential elements, are considered to be co-absorbed by the mycelium alongside Ca and accumulation patterns in mushrooms more or less follow the natural pattern in soil substrata. Thus, some of the detectable differences in the normalized pattern could potentially be reflecting the local soil for a given REE(s).

For the REEY (Figure 2 and Appendix A)-normalized patterns (caps, stipes, whole *B. edulis,* and including the literature data), Y showed a small positive deviation in all the plots which in general exhibited an almost flat, unchanging profile. A similar decoupling was observed in studies of mushrooms *M. procera* and *Suillus luteus* [22,24]. A strong positive Eu anomaly and abundance of REE were found in *B. edulis* from the Augustowska Primeval Forest (Appendix A; Table 1) and this can be related to specific nature of the site as observed for *M. procera* [24]. A positive Eu anomaly was also reported in some vegetation (moss) from south–central Poland [43].

### 3.4. REE Intakes from Bolete Meals

The sum of REEs (∑REE) in the King Bolete varied from 86.7–758 µg kg^−1^ dw (median 310 µg kg^−1^ dw; mean ± SD, 327 ± 186 µg kg^−1^ dw; for conversion to wet weight a divider of 10 can be used assuming moisture content in this species at 90%) for the whole fruiting bodies (Table 1) excluding the very high value of 1796 µg kg^−1^ dw for the Warmia—Morąg site. Because of the high gourmet value of the King Bolete and the rather unique data on REEs obtained in this study for a relatively large collection of this mushroom, an attempt was made to estimate possible intake of these nonessential elements from theoretical bolete meals. Considering the median ∑REE of 0.031 mg kg^−1^ wet (whole) weight (0.31 mg kg^−1^ dw; assuming 90% of moisture) in mushrooms (whole fruiting bodies), a meal of 100 g (whole weight) at one time would expose a consumer to 0.0031 mg of REEs, and consumption of 300 g by high level consumers would result in an intake of 0.0093 mg of REEs. The consumption of such mushroom meals daily for a week would result in the intake of 0.0217 and 0.0651 mg, respectively. Considering the highest ∑REE value of 758 µg kg^−1^ dw (0.0758 mg kg^−1^ ww) observed for the Augustów Primeval Forest site (id 6), the consumption of 100 g and a 300 g mushroom meals would respectively result in daily intakes of 0.0076 mg and 0.0227 mg of REEs. Weekly, the consumption of 100 g and 300 g meals of King Bolete collected at Augustów Primeval Forest site would result in intakes of 0.0531 mg and 0.159 mg, respectively.

The effect of culinary processing of mushrooms, including the King Bolete, on the fate of the REEs is unknown. It is anticipated that, as with other elements for which there are data, that depending on a culinary treatment, the concentration in mushroom meals may be smaller, similar, or slightly greater than in the raw unprocessed (wet, whole weight) product. Thus, the assessment is made on the basis of data for the raw (unprocessed) fruiting bodies. The estimated dietary intake (EDI) of REEs from the consumption of King Bolete for an adult of weight 60 kg ranges from 0.014 to 0.126 µg kg^−1^ day^−1^ for consumption of 100 g mushroom meal and from 0.043–0.379 µg kg^−1^ day^−1^ for consumption of a 300 g mushroom meal. Even for the Warmia–Morąg site, with a highly elevated ∑REE value of 179.6 µg kg^−1^ ww, the EDI would be 0.299 µg kg^−1^ day^−1^ for 100 g and 0.898 µg kg^−1^ day^−1^ for consumption of 300 g mushroom (whole weight) meal. These intakes are much lower than the EDI reported in the literature that would be damaging to human health (100–110 µg kg^−1^ day^−1^) [3,45,46]. Considering the median ∑REE of 31 µg kg^−1^ ww for whole fruiting bodies, a consumer would need to eat about a 100 g mushroom meal (EDI of 0.0516 µg kg^−1^ day^−1^) daily for about 1953 days (279 weeks) to exceed the EDI threshold (100 µg kg^−1^ day^−1^), while for a child this amounts to about 970 days (138 weeks) (EDI of 0.103 µg kg^−1^ day^−1^).

For a child of weight 30 kg that is able to consume a similar amount as an adult, the EDI ranges from 0.029 to 0.253 µg kg^−1^ day^−1^ for consumption of a 100 g mushroom meal and 0.087–0.758 µg kg^−1^ day^−1^ for consumption of a 300 g mushroom meal. Moreover, for children consuming boletes from the Warmia–Morąg site, the EDI would be 0.599 µg kg^−1^ day^−1^ for a 100 g mushroom meal and 1.796 µg kg^−1^ day^−1^ for consumption of a 300 g mushroom meal. Therefore, the health risk assessment of intake of the studied King Bolete shows that the mushroom presents no health risks from REEs, though it would be unusual to see an adult eat as much as a 300 g quantity, even during the peak of the mushroom picking season. Culinary processing of mushrooms (including blanching, boiling, blanching, and pickling) lowers the amount of the metallic elements in mushroom meals (wet weight basis) [56,57,58], while frying or braising to some degree prevents loss or slightly enhances their quantity on a wet (whole) weight basis, as has been observed in the case of Hg, ^137^Cs, and ^40^K [59,60,61]. Even after consumption, intestinal absorption of metallic elements from a mushroom (culinary processed or only dried) is generally considered to be below 50 % [58].

## 4. Conclusions

The literature on REE in foodstuffs, and mushrooms in particular, is scarce. This study documented the rare-earth element contents of the popular and valued edible mushroom, the King Bolete (*B. edulis*), foraged across Poland. Results of this study show rather low contents of REEs in the whole mushroom and in the separate morphological parts (caps and stipes). Available data also show that the possibility of contamination of the fruiting bodies of *B. edulis* by rare-earth elements from their increased usage worldwide in the last decades seems unlikely. Thus, from the point of view of REE contamination, intake of this mushroom, even for high level consumers, would not present any known toxicological risks.

## Figures and Tables

**Figure 1 ijerph-19-08948-f001:**
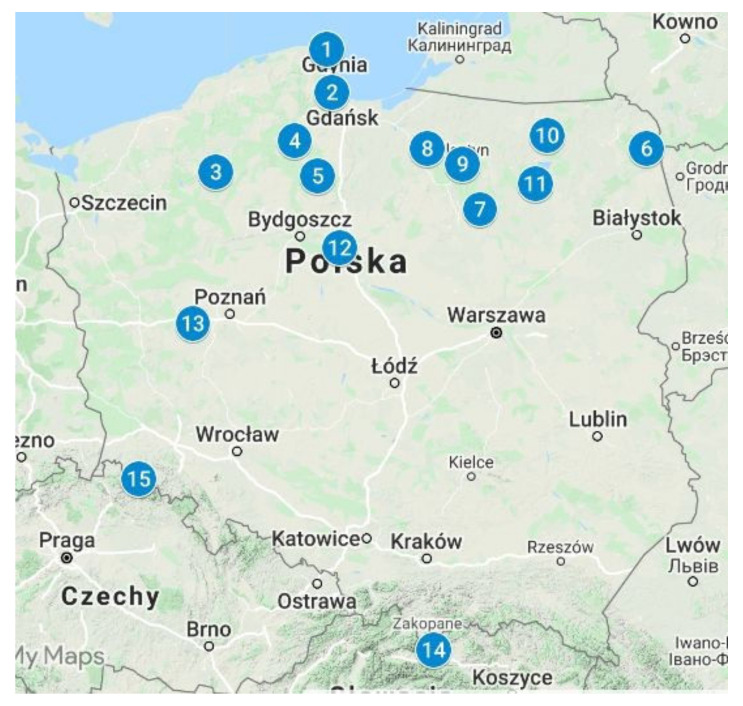
Sampling locations of King Bolete [(id1) Seacoast Landscape Park; (id2) Tricity Landscape Park, Osowa; (id3) Pomerania, Szczecinek; (id4) Wdzydze Landscape Park; (id5) Tuchola Pinewoods; (id6) Augustów Primeval Forest; (id7) Warmia land, Puchałowo; (id8) Warmia, Morąg; (id9) Warmia, Olsztyn; (id10) Mazury, Giżycko; (id11) Mazury, Piska Forest; (id12) Kujawy, Toruń forests; (id13) Greater Poland, Porażyn; (id14) Tatra Mountains; (id15) Sudety Mountains, Kłodzka Dale] (see also Table 1).

**Figure 2 ijerph-19-08948-f002:**
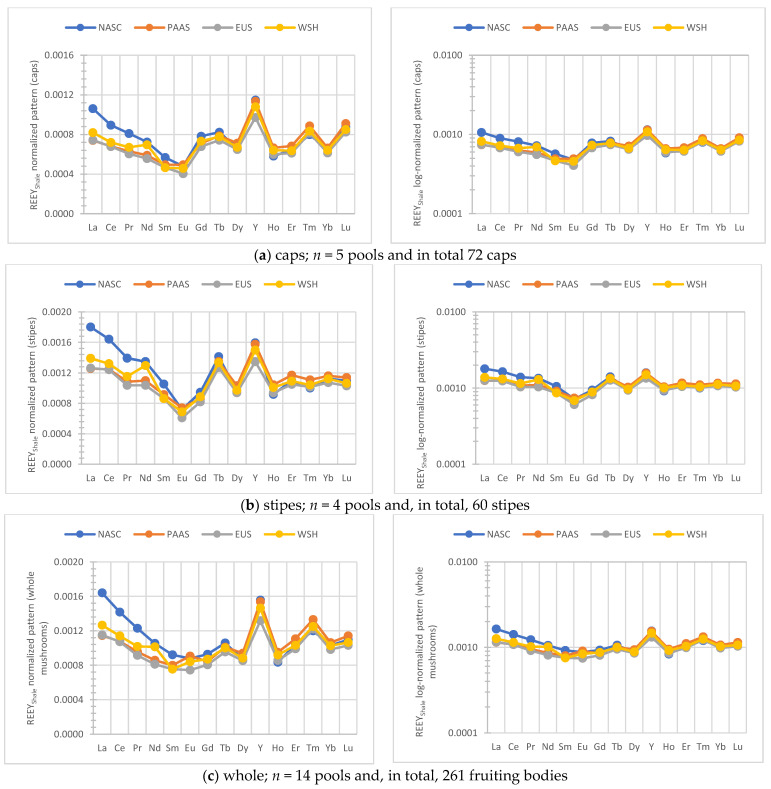
The shale (NASC, PAAS, EUS and WSH) normalized normal and log-normal patterns of REEY in *B. edulis* (medians for the caps, stipes, and the whole fruiting bodies).

## Data Availability

Not applicable.

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
