# Peer review of "Rare Earth Elements in Boletus edulis (King Bolete) Mushrooms from Lowland and Montane Areas in Poland"

_ijerph, 2022, doi:10.3390/ijerph19158948_

Round 1
Reviewer 1 Report
Rare Earth Elements in Boletus edulis (king bolete) Mushrooms from Lowland and Montane Areas in Poland
By Jerzy Falandysz
General Comments
The paper presents the concentration of REEs in samples from B. edulis collected in Poland. There are not so much data on REEs in this matrix and in a world where the use of REEs is increasing it is relevant to have this information. Actually, mushrooms are not essential in the human diet the pollutants contained in them can undergo bioaccumulation processes in the food chain. For this reason, the paper even if it is rather descriptive, deserve the publication on IJERPH. Here below some suggestions to improve the results presentation and discussion.
Specific comments
The acronymous have to be made explicit the first time they appear in the text. Abstract and Graphical abstract have to be self-consisting, therefore the acronymous have to specified there (REEY, NASC, PAAS, EUS, WSH)
Materials and Methods
Line 5 What does “The climate is moderate and changing” mean? The authors are referring to temperatures or to meteorological conditions? Remember that speaking of climate means that you have the variability of meteorological conditions at least for 30 years. Please clarify the sentence.
REE Absolute Concentration Levels
The sentence “The median values of the individual elements derived for whole fruiting bodies, caps and stipes in this study were in the range (in μg kg‐1 dw), 5.9 to 35 for Sc, 31 to 43 for Y, 33 to 56 for La, 60 to 110 for Ce, 6.4 to 11 for Pr, 22 to 41 for Nd, 3.4 to 6.3 for Sm, 0.6 to 1.1 for Eu, 4.3 to 5.2 for Gd, 0.7 to 1.2 for Tb, 3.8 to 5.5 for Dy, 0.7 to 1.1 for Ho, 2.1 to 3.6 for Er, 0.4 to 0.6 for Tm, 2.0 to 3.5 for Yb and 0.4 to 0.5 for Lu (Table 1)” is not useful, the readers can find these value in the table 1.
How the bioconcentration factor (BCF) is calculated? BCF >1 means that Ca concentrations are lower than in the soil? What soil is taken as reference: the mean upper continental crust or more specific soil where mushrooms live? Please specify and discuss this in the text as this topic is particularly relevant to understand the potentiality of mushrooms to act as bioaccumulators of REE.
“The light REEs (LREE; here: La, Ce, Pr and Nd) occurred in mushrooms in higher concentrations when compared to the middle REEs (Sm to Dy) and heavy REEs (from Ho to Lu) (Table 1)”: this is the patter of the abundance of REE on the Earth, please discuss the are differences between the ratio LREE/HREE in mushrooms and in the soil in particular from Europe (EUS).
Shale Normalized Patterns of REEY in B. edulis
The information of log-normalized pattern is redundant, the variations are not in the order of magnitude, therefore log-normal plots are not useful. The pattern is better visible in the simply normalized plot. I suggest deleting the all the log normal plots (also in the graphical abstract).
It is not clear the reason for normalization with shale from different mainland if mushrooms are collected in Europe. What is the conclusion that the authors want to obtain from the normalization with shale from different soil? To understand if there is a particular uptake for one of the REE respect to the soil the normalization have to be made on European soil.
Some of the values in the table and plots reported in the supplementary information should be moved in the main text as they are important to understand if anomalies in the normalized profile of REE or REEY are specific for the species B. edulis or for the sampling site.
Data from Bau et al., 2018 reported in the supplementary information are related to mushroom collected in Europe (a least it seems from the title of publication) but from where in Europe? This information has to be reported. Besides. the data from Bau et al., 2018 are from specifical part of the mushroom (carpophore), could be this the reason for the different pattern of normalized profile of REE, excluding Y anomaly that have the same pattern in all the samples?
Author Response
Reviewer 1
General Comments
The paper presents the concentration of REEs in samples from B. edulis collected in Poland. There are not so much data on REEs in this matrix and in a world where the use of REEs is increasing it is relevant to have this information. Actually, mushrooms are not essential in the human diet the pollutants contained in them can undergo bioaccumulation processes in the food chain. For this reason, the paper even if it is rather descriptive, deserve the publication on IJERPH. Here below some suggestions to improve the results presentation and discussion.
Specific comments
1Q.The acronymous have to be made explicit the first time they appear in the text. Abstract and Graphical abstract have to be self-consisting, therefore the acronymous have to specified there (REEY, NASC, PAAS, EUS, WSH)
R: Agreed.
Materials and Methods
2Q.Line 5 What does “The climate is moderate and changing” mean? The authors are referring to temperatures or to meteorological conditions? Remember that speaking of climate means that you have the variability of meteorological conditions at least for 30 years. Please clarify the sentence.
R: Is amended.
Poland has a largely flat landscape with uplands in the south, and agricultural land and forests dominate, and has four seasons - spring, summer, autumn and winter. The climate (temperature and meteorological conditions) is moderate and changing and becomes more continental with increasing distance from the sea to the montane south of the country.
Note: “Poland has a moderate climate with both maritime and continental elements. You can count on many sunny days and many rainy days and don't be surprised when the summer turns out to be quite hot or quite rainy. Winters are usually cold, with temperatures well below freezing, and more or less snowy.” History of meteorological conditions monitoring in Poland is much over 30 years.
REE Absolute Concentration Levels
3Q The sentence “The median values of the individual elements derived for whole fruiting bodies, caps and stipes in this study were in the range (in μg kg‐1 dw), 5.9 to 35 for Sc, 31 to 43 for Y, 33 to 56 for La, 60 to 110 for Ce, 6.4 to 11 for Pr, 22 to 41 for Nd, 3.4 to 6.3 for Sm, 0.6 to 1.1 for Eu, 4.3 to 5.2 for Gd, 0.7 to 1.2 for Tb, 3.8 to 5.5 for Dy, 0.7 to 1.1 for Ho, 2.1 to 3.6 for Er, 0.4 to 0.6 for Tm, 2.0 to 3.5 for Yb and 0.4 to 0.5 for Lu (Table 1)” is not useful, the readers can find these value in the table 1.
R: Is hard to agree with this comment. In the sentence above are given the ranges of median values (two values only) for three morphological parts of mushroom. So, we would like to keep this as it is. In Table 1 these parameters are separated (with blocks of individual, mean, uncertainty and median data) for each morphological part, so, for a more deeper insight.
4Q How the bioconcentration factor (BCF) is calculated?
- BCF alias “quotient”.
BCF >1 means that Ca concentrations are lower than in the soil?
- Don’t understand. If Ca in mushroom is lower than in soil how this (BCF alias quotient) could be “> 1”?
What soil is taken as reference: the mean upper continental crust or more specific soil where mushrooms live? Please specify and discuss this in the text as this topic is particularly relevant to understand the potentiality of mushrooms to act as bioaccumulators of REE.
- This part of discussion is about the Ca in mushrooms (Boletus edulis) and adjacent forest topsoils (0-10 cm layer) collated from other studies. Mushrooms don’t act as bioconcentrators (bioaccumulators) [bioconcentration and bioaccumulation are two different terms]. Mushrooms accumulate some Ca and REEs but they don’t bioconcentrate them (BCF for REEs is well below 1) – is much more of REEs in forest topsoil than in emerging fruiting bodies (see also in the articles by Bau et al. 2018; Grawunder and Gube, 2018; Vukojević et al., 2019; Zocher et al. 2018).
4Q “The light REEs (LREE; here: La, Ce, Pr and Nd) occurred in mushrooms in higher concentrations when compared to the middle REEs (Sm to Dy) and heavy REEs (from Ho to Lu) (Table 1)”: this is the patter of the abundance of REE on the Earth, please discuss the are differences between the ratio LREE/HREE in mushrooms and in the soil in particular from Europe (EUS).
- This is good question but probably for a review paper (if more good data will be available – published). At the moment we have only 1 (one) good data (CREDIBLE RESULT) available for a single fruiting body of B. edulis and 3 samples of whole fruiting bodies of Suillus lueteus (all from Germany). This can be (, please discuss the are differences between the ratio LREE/HREE in mushrooms and in the soil in particular from Europe (EUS) question of quality of analytical data published. Thus, this is a somehow outside of the scope of our manuscript – we don’t attempt in this manuscript (discussion) to elaborate any comments (critical) about some other published articles (which are intentionally not cited in this manuscript because in our opinion they are with evidently biased results).
Shale Normalized Patterns of REEY in B. edulis
5Q The information of log-normalized pattern is redundant, the variations are not in the order of magnitude, therefore log-normal plots are not useful. The pattern is better visible in the simply normalized plot. I suggest deleting the all the log normal plots (also in the graphical abstract).
- We disagree with this comment. However from other point of view (see below) this comment is or can be useful for other type of study.
The log-normalized patters are very helpful (see also discussion in Bau et al. 2018 and Zocher et al. 2018). The log-normalized patterns are very helpful not only to detect natural anomaly (anomalies) in fractionation of the REEs but are also good proof for detection of the incredible (false) results due to analytical bias. So, they are very helpful from both sites. The simply normalized “zigzag” plot has a value also but from other point of view – we don’t want to discuss this in our original manuscript because this kind of discussion is good for a review or commentary (critical comment) letter, which is not the aim of this study. Because we want to keep the log-normalized patterns (reason indicated) – we don’t want to double the Figures with extra “zigzag” patterns – what would be immediately criticized as “doubling” of the figures.
6Q It is not clear the reason for normalization with shale from different mainland if mushrooms are collected in Europe. What is the conclusion that the authors want to obtain from the normalization with shale from different soil? To understand if there is a particular uptake for one of the REE respect to the soil the normalization have to be made on European soil.
- Hard to agree with this comment. A reason and the results are clear. The mushrooms, soils/forest topsoils and shales – regardless of a place of origin – reflect the same REE distribution pattern – they reflects the Oddo–Harkins rule. If not this will simply show on biased results. The attempt to normalize patterns using also the European shale is just to confirm that practically is no difference with other shales (and soils, eg. “mean“, the European or world soil.
7Q Some of the values in the table and plots reported in the supplementary information should be moved in the main text as they are important to understand if anomalies in the normalized profile of REE or REEY are specific for the species B. edulis or for the sampling site.
R: We are begging for mercy. We already have “6” drawings – figures in One (full page). Supplementary Material is provided for possible reader with a willing for a deeper insight, so, in ready form and no need to calculate/plot himself. We (as stated in manuscript) also want to be careful when discussing anomalies we found for some sites (samples).
8Q Data from Bau et al., 2018 reported in the supplementary information are related to mushroom collected in Europe (a least it seems from the title of publication) but from where in Europe? This information has to be reported. Besides. the data from Bau et al., 2018 are from specifical part of the mushroom (carpophore), could be this the reason for the different pattern of normalized profile of REE, excluding Y anomaly that have the same pattern in all the samples?
R: Agreed. Is added (collected in Germany) and (from Germany) – they don’t specify where in Germany their B. edulis was collected. Carpophore is in other words a fruiting body or fruitbody or sporocarp or …– mycologist use different words for the same. If we understand this correctly - from our point of view or from what we know now, the quality of analyses and lack of by-side contamination (sand/soil) are major factors and for carpophores (whole fruiting bodies) patterns are similar for various macrofungi.
Reviewer 2 Report
1.First sentence of abstract should be rephrased.
2.Permissible limits of REE for consumers need to be specified.
3.If REE concentrations in morphological parts of the fruiting bodies of B. edulis are available, same may be clearly described.
4.Is there any linkage of REE enrichment with soil types?
5.In studied materials, what are the processes involved in REE enrichment?
6.Levels of REE concentrations in different types of mushrooms need to be included.
7.It is necessary to state the values typical of various species.
8.Usage of word “Absolute” for concentration should be avoided.
9.REE concentration in studied samples should be expressed in parts per million (PPM).
10.In each row (Table 1) type of mushrooms studied may also be included.
Author Response
Reviewer 2
Comments and Suggestions for Authors
1.First sentence of abstract should be rephrased.
R. Is amended.
Mining / exploitation and commercial applications of the rare earth elements (REEs: La, Ce, Pr, Nd, Sm, Eu, Gd, Tb, Dy, Ho, Er, Tm, Yb and Lu) in the past 3 decades have raised concerns about their emissions to the environment, possible accumulation in food webs, and occupational / environ-mental health effects.
2.Permissible limits of REE for consumers need to be specified.
R: What kind of “permissible limits”? In the European Union is no “permissible limits” for the REEs in foods. They are not normalized and not considered as essential or nonessential – probably because of extremely low concentrations in staple foods.
3.If REE concentrations in morphological parts of the fruiting bodies of B. edulis are available, same may be clearly described.
Jeśli dostępne są stężenia REE w częściach morfologicznych owocników B. edulis, można to jasno opisać.
4.Is there any linkage of REE enrichment with soil types?
- NOT for forest mushrooms.
5.In studied materials, what are the processes involved in REE enrichment? - Do you mean uptake by fungal mycelium? REEs are not essential elements for fungi (and man) – so process is always passive (co-absorption with some essential elements, e.g. calcium and maybe also magnesium).
6.Levels of REE concentrations in different types of mushrooms need to be included. - Levels of REE concentration (from credible studies) are already included in section on Discussion.
7.It is necessary to state the values typical of various species. - ?. What do you mean “various species” ? Other than B. edulis (this study).
8.Usage of word “Absolute” for concentration should be avoided.
- Ok. Is: 3.1. REE concentration levels
Measurement of Absolute Concentration at the Subcellular Scale by Brittney L. Gorman, Melanie A. Brunet, Susan N. Pham, and Mary L. Kraft*
In this Perspective, we discuss a significant advance in using nano secondary ion mass spectrometry (nanoSIMS) to measure the absolute concentration of a 13C-labeled metabolite within secretory vesicles, as reported by Thomen et al. in the April issue of ACS Nano.
9.REE concentration in studied samples should be expressed in parts per million (PPM).
- R. In our opinion, the PPM values are good for the REEs in soil, shales, rocks, sediments… but not for the biological samples such as mushrooms, foods etc. This is because the REEs in mushrooms and other biological materials are at the PPB level, so, use of a such unit as “μg kg-1 dw” for mushroom is credible.
In each row (Table 1) type of mushrooms studied may also be included. - R. What do you mean?
We have only one species (and so one type .. mycorrhizal mushroom) .. as stated in the body text of the manuscript:
Boletus edulis Bull., is an indigenous and edible mycorrhizal mushroom (King Bolete), ..
Reviewer 3 Report
This work is potentially interesting, but needs a major revision, with a reflection on the methodology used.
1. The sampling period raises doubts about its impact on the final data provided. Over a decade, soils may have changed significantly.
2. 2. The use of geological patterns to denote rare earth enrichments or depressions is questionable, as these are data obtained from ancient rocks. It would be desirable to adopt a regional background, meaning the minimum rare earth value in an area, and apply it to obtain possible enrichments. It would be equivalent to the geological geoaccumulation index.
3. Since higher concentrations of rare earths are attributed to previous military use of certain areas, it would be advisable to have data on the rare earth content of the soils on which the Boletus specimens were obtained.
4. The discussion should be improved, with comparison in tables of the results obtained with other similar papers at the Polish or European level.
5. Some tables should be added and Figure 1 should be improved.
6. Some additional comments have been included in the attached pdf.

Author Response
Reviewer 3
Comments and Suggestions for Authors
This work is potentially interesting, but needs a major revision, with a reflection on the methodology used.
- The sampling period raises doubts about its impact on the final data provided. Over a decade, soils may have changed significantly.
- There is no any evidence that “over a decade” forest soils at the sites sampled “have changed significantly” – and especially as regard to the REEs. We don’t understand how “The sampling period could raise doubts about its impact on the final data provided”?
- The use of geological patterns to denote rare earth enrichments or depressions is questionable, as these are data obtained from ancient rocks. It would be desirable to adopt a regional background, meaning the minimum rare earth value in an area, and apply it to obtain possible enrichments. It would be equivalent to the geological geoaccumulation index.
- The use of shales for REEs normalized patterns is so far accepted and practiced. See at very good article : Uranium, thorium and rare earth elements in macrofungi: what are the genuine concentrations? By Jan Borovicˇka • Jaroslava Kubrova´ , Jan Rohovec • Zdeneˇk Rˇ anda • Colin E. Dunn. Of course also use of ”a regional background” can be good too but we don’t have soil data (REEs) for our mushrooms. On the other side, also in the very good article (high quality analytics) “Element distribution in fruiting bodies of Lactarius pubescens with focus on rare earth elements” by Anja Grawunder and Matthias Gube” a shale is used for normalisation (PAAS according to McLennan (1989). So, is acceptable and even recommended.
- Since higher concentrations of rare earths are attributed to previous military use of certain areas, it would be advisable to have data on the rare earth content of the soils on which the Boletus specimens were obtained.
- Yes, but we don’t have results for REEs in soils (which levels are three to four or more orders of magnitude richer in the REEs than mushrooms. We have to be careful with discussion and explanation also because according to some authors, stipe of some mushrooms (specimens) can be incrusted with particles of sand – which can’t be removed by brushing /washing of a mushroom.
- The discussion should be improved, with comparison in tables of the results obtained with other similar papers at the Polish or European level.
- In discussion are cited in practice all credible results from other authors (only a few such articles published). Your suggestion is good for the REVIEW paper but this can be hardy painful – see also in the already mentioned article: Uranium, thorium and rare earth elements in macrofungi: what are the genuine concentrations? By Jan Borovicˇka • J.
- Some tables should be added and Figure 1 should be improved.
- What kind of Tables? This is original work with one large Table 1. Two extra Tables are included in the Supplementary material. We are begging for mercy. This figure is for reference only and is quite OK - no one other wants to replace it. Information about the site names (and adjacent forest - if applicable) is provided in the Table.
- Some additional comments have been included in the attached pdf.
Comments made by Reviewer 3 directly in manuscript body text
- The collection period covers a decade, which represents a methodological problem as changes in soil or climatic conditions may have occurred. It would have been more appropriate to concentrate the collection in a single year.
Repetition of the question 1.
- There is no any evidence that “over a decade” forest soils at the sites sampled “have changed significantly” – and especially as regard to the REEs. We don’t understand how “The sampling period could raise doubts about its impact on the final data provided”?
- As the study area is very large, some reference should be made to the climate and soil type at each sampling point. This could be included in a new table.
- ?. Regarding to climate – as amended: Poland has a largely flat landscape with uplands in the south, and agricultural land and forests dominate, and has four seasons - spring, summer, autumn and winter. The climate (temperature and meteorological conditions) is moderate and changing and becomes more continental with increasing distance from the sea to the montane south of the country.
There is no any significant difference in climate (weather) in Poland (area is only 312,000 square kilometres) on the annual basis and over time – is very similar (mean).
Regarding soil types ... Soil bedrock in Poland is highly diverse and is reflected in the soil types with a pre-dominance of brown earth (brown and lessive soils), podzols (podzolic soils, podzols and rusty soils) and several other types [37]. There is no a wider information that soil type has effect on uptake of REEs by fungal mycelium and accumulation in Fruiting bodies.
3.This figure needs to be changed and improved. Some locations overlap with city or country names.
R: We are begging for mercy. This figure is for reference only and is quite OK - no one other wants to replace it. Information about the site names (and adjacent forest - if applicable) is provided in the Table.
Q4. I have doubts about the use of geological patterns to analyse REE patterns in B. edulis. An additional possibility would be to obtain a regional background of REE in B. edulis and apply some enrichment index. The authors should consider alternative methodologies for measuring REE enrichment.
- R. There is no certified (mushroom) reference materials for fungi. These Chinese [citrus leaf (GBW 10020) and soil (GBW 07405) CRMs] materials are very well certified for REEs content. Since analyses were made in geo-lab in China (during our stay there) as stated in acknowledgements,
- This paragraph should be included in the introduction, as it adds nothing to the discussion.
- This is here because of relation to the REEs, which are not essential for mushrooms – possible co-absorption with essential elements (Ca, Mg).
6Q. Genus and species names should be in italics.
R: Off course it should be. In original manuscript as submitted this is in italics (we don’t know why after a file submission this (all wards in Latin) was changed by the Editorial Office server to the non-italics?).
- Reference?
R: Is added. [48]
8Q. These paragraphs should be summarised and accompanied by a new table.
R: Is added: The effect of culinary processing of mushrooms, including the King Bolete, on fate of the REEs is unknown. This is anticipated that, as is with other elements for which are data, that depending on a culinary treatment, the concentration in mushroom meal may be smaller, similar or slightly greater than in the raw unprocessed (wet, whole weight) product. Thus, the assessment is made on the basis of data for the raw (unprocessed) fruiting bodies. The estimated dietary intake (EDI) of REEs from the consumption of King Bolete for an adult of weight 60 kg ranges from 0.014 to 0.126 μg kg‐1 day‐1 for consumption of 100 g mushroom meal and from 0.043 – 0.379 μg kg‐1 day‐1 for consumption of a 300 g mushroom
meal.
Round 2
Reviewer 1 Report
Dear Author,
I understand that this not a review paper, but I give only few hints to improve the discussion. I respect your opinion that you disagree with me, and the paper can be published, but it still seems a data presentation report more than a scientific publication.
Reviewer 3 Report
The revision has been adequate and the manuscript can be accepted.